# Gelatin/Hyaluronic Acid Scaffold Coupled to CpG and MAGE-A5 as a Treatment against Murine Melanoma

**DOI:** 10.3390/polym14214608

**Published:** 2022-10-30

**Authors:** Gabriela Piñón-Zárate, Beatriz Hernández-Téllez, Katia Jarquín-Yáñez, Miguel Ángel Herrera-Enríquez, América Eréndira Fuerte-Pérez, Esther Alejandra Valencia-Escamilla, Andrés Eliú Castell-Rodríguez

**Affiliations:** Facultad de Medicina, National Autonomous University of Mexico, Ciudad de México 04510, Mexico

**Keywords:** gelatine/hyaluronic acid scaffolds, immunomatherials, immunotherapy, melanoma

## Abstract

The half-time of cells and molecules used in immunotherapy is limited. Scaffolds-based immunotherapy against cancer may increase the half-life of the molecules and also support the migration and activation of leukocytes in situ. For this purpose, the use of gelatin (Ge)/hyaluronic acid (HA) scaffolds coupled to CpG and the tumor antigen MAGE-A5 is proposed. Ge and HA are components of the extracellular matrix that stimulate cell adhesion and activation of leucocytes; CpG can promote dendritic cell maturation, and MAGE-A5 a specific antitumor response. C57BL/6 mice were treated with Ge/HA/scaffolds coupled to MAGE-A5 and/or CpG and then challenged with the B16-F10 melanoma cell line. Survival, tumor growth rate and the immune response induced by the scaffolds were analyzed. Ge/HA/CpG and Ge/HA/MAGE-A5 mediated dendritic cell maturation and macrophage activation, increased survival, and decreased the tumor growth rate and a tumor parenchyma with abundant cell death areas and abundant tumor cells with melanin granules. Only the scaffolds coupled to MAGE-A5 induced the activation of CD8 T cells. In conclusion, Ge/HA scaffolds coupled to CpG or MAGE-A5, but not the mixture, can induce a successful immune response capable of promoting tumor cell clearance and increased survival.

## 1. Introduction

Cancer immunotherapy is a therapeutic strategy based on the activation of the immune system to induce the elimination of tumor cells [1,2]. Tumor antigens, immunomodulatory molecules, cytokines, antibodies, and cells have been used in immunotherapy; however, there are still patients who do not respond to therapies, showing a high tumor growth rate and low survival [3]. There are several reasons for the failures in classical immunotherapy, such as the decrease in the half-life of immunomodulatory molecules, antibodies, and cells; the inhibitory microenvironment induced by the tumor; or the poor lymphocyte migration toward the tumor parenchyma [4]. The use of implantable scaffolds has become an alternative in immunotherapy since the biomaterials used in their construction can be coupled to antibodies, antigens, immunomodulators, cytokines or chemokines, which can be released little by little, provoking multiple effects, such as preferential cell migration, adhesion, and activation of leukocytes [5,6]. Therefore, in experimental antitumor immunotherapy, scaffolds constructed with biomaterials such as silica, collagen, alginate, fibrin and glycosaminoglycans have promoted the development of successful antitumor immune responses that helped to increase survival and decrease the rate of tumor growth compared to the results obtained in classical immunotherapy [2,4,7,8].

In addition, biomaterials may be found in the extracellular matrix and be recognized by cells of the immune system, mainly antigen-presenting cells. Gelatin (Ge) and hyaluronic acid (HA) are biomaterials abundant in connective tissue that are capable of inducing the migration, adhesion, and activation of leukocytes [9,10,11,12]. Ge is the product of collagen hydrolysis and is known to possess RGD domains involved in the adhesion and recognition of scaffolds by fibroblasts, macrophages, neutrophils, and even lymphocytes; in addition, the implantation of collagen or gelatin can induce the production of TNGα and IL-6 [10,12]. In contrast, HA is a glycosaminoglycan commonly found from the dermis to the mucosal lamina propria. HA has characteristics that enable the formation of a microenvironment necessary for cell survival, since HA may bind to growth factors; also, macrophages, dendritic cells (DC), and lymphocytes may recognize HA through the CD44 receptor, promoting cell migration; in some cases, when structural changes occur in HA, DC and macrophages are activated after recognition as a tissue damage-associated molecule (DAMP) [11]. In addition, macrophages and DC can identify HA through Toll-like receptors (TLRs) 2 and 4, which promote its activation by initiating an increase in the expression of costimulatory molecules and the production of IL-12 [9,13,14]. Therefore, a scaffold constructed with Ge and HA would promote leukocyte migration, adhesion, and activation, helping to coordinate the development of a prolonged, successful, and specific antitumor response. In addition, for the development of a successful immune response, it is necessary to couple immunomodulatory molecules to the scaffold to allow the correct activation of leucocytes.

Among immunomodulatory molecules, pathogen-associated molecular patterns (PAMPs) are known to stimulate the immune system. Bacterial DNA is a PAMP recognized by TLRs, especially when unmethylated CpG sequences are found. Synthetic oligodeoxynucleotides (ODNs) contain CpG recognized by TLR9 in DC and macrophages, supporting the expression of costimulatory molecules, the production of type I IFN, the development of a Th1 response, and the indirect activation of NK cells [15,16,17]. However, tumor antigens are known to lead to specific immune responses that promote the elimination of tumor cells. MAGE proteins are some of the tumor antigens widely used in classical immunotherapy. MAGE genes are located on the X chromosome and may be expressed in melanoma cells, bladder cancer, brain, prostate, ovary, skin, and thyroid [18]. In addition, MAGE antigens are presented via MHCI and II to T lymphocytes, so they can provoke the development of specific immune responses when they have been used as a complement in classical immunotherapy based on the use of DC, lymphocytes or viruses [19,20,21].

Thus, scaffolds built with extracellular matrix biomaterials can be an alternative to improve the results obtained by classical cancer immunotherapy [22,23], since they can protect immunomodulatory molecules from degradation and allow their gradual release. In this sense, CpG and MAGE antigens coupled to a Ge and HA scaffold will likely trigger a robust antitumor immune response. Thus, the objective of this study was to evaluate this immune response in a murine melanoma model.

## 2. Materials and Methods

### 2.1. Mice

For the in vivo experiments, six male C57BL/6 mice of 6 to 8 weeks of age with the genotype H2Kb were used. To calculate the sample size, an equation based on the assessment of incidences was used:X = N ((A/100) (B/100) ((C) (100))
where: X = Final number of animals needed; N = is the minimum statistical number needed to conclude the project objectives; A = 100 - incidence 1; B = 100 - % incidence 2 and C = 100 - % incidence 3 and so on.

In our project we have the following data:

N = at least 5 mice according to various articles in which melanoma induction is performed.

A = 10. Probability of death of a mouse not attributable to the procedure.

B = 10. Probability of death due to some complication in the procedure.

Substituting values:

X = 5/ ((90/100) (90/100)) = 6.17 mice. Therefore, the model is adjusted to 6 mice per group.

Mice were kept under light-dark, temperature-controlled conditions and fed ad libitum in the Animal Facility of the Department of Cell Biology of the Faculty of Medicine, UNAM. A maximum of four mice were kept in clean and dry conditions in boxes with solid and continuous floors and walls with a removable grid lid, with rounded edges and containing sawdust beds. Food and drinking water were freely available. All the above were performed according to the Official Mexican Standard of technical specifications for the production, care, and use of laboratory animals (NOM-062-ZOO-1999).

### 2.2. Ethics Statement

The study was approved by the School of Medicine’s Ethics and Research Committees (Comisiones de Investigación y de Ética, dictamen 141/2015), Universidad Nacional Autónoma de México. This study was performed in accordance with the Official Mexican Standard NOM 062-ZOO-1999

### 2.3. Reagents

The monoclonal antibodies for the characterization of DC and lymphocytes by flow cytometry were the following: anti-CD3 FITC, anti-CD4 APC, anti-CD8 PE, anti-CD137-biotin, anti-CD11c APC, anti-CD40-PE, anti-CD86-PE, anti-CD80-PE, anti-F4/80-Cy5 and anti-Ia/Ie-FITC from Biolegend (San Diego, CA, USA). The murine melanoma cell line B16-F10 with haplotype H-2Kb was purchased from The American Type Culture Collection, USA. RPMI-1640 culture medium was purchased from Biowest (Nuaille, France). Ge from bovine skin with a protein percentage of 78% and a water content of 8% was used. HA sodium salt from *Streptococcus equi*, both from SIGMA (St Louis, MO, USA) with a solubility of 5 mg/mL and an MW of ~1.5–1.8 × 10^6^ Da was used.

### 2.4. Tumor Antigen

The MAGE-A5 peptide (LGITYDGM) was synthesized by Research Genetics (Invitrogen, Leiden, The Netherlands) with a purity of 94%.

### 2.5. Ge and HA Scaffold Construction

Ge/HA was dissolved in distilled water at a ratio of 4:1 under constant stirring (IKA C-Mag HS7) at 750 rpm for 30 min at 50 °C. One milliliter of the Ge/HA solution was placed in 1 mL conical tubes. In the case of the Ge/HA scaffolds coupled with CpG or MAGE-A5, 25 mM MAGE-A5 and 10 µg CpG were added to the Ge/HA mixture, according to [24,25]. Subsequently, the mixtures were frozen with liquid nitrogen and then lyophilized for 12 h at −50 to 0.036 mbar of pressure. The scaffolds obtained were immersed in 50 mM dissolved 96% for 24 h at 4 °C. Then, the scaffolds were washed 3 times with distilled water and lyophilized again under the same conditions mentioned above. The scaffolds obtained were sectioned into disks approximately 9 mm in diameter by 3 mm thick. The scaffolds were then pulverized with a Ultraturrax T25 IKA (Northchase Parkway SE, Wilmington, NC, USA) at 12,200 rpm for 1 min and hydrated with 500 µl of PBS to form a hydrogel.

### 2.6. Fourier Transform Infrared (FT-IR) Characterization

The chemical composition of Ge/HA polymers crosslinked with and without EDC was assessed by using a spectrometer (NICOLET Nexus 670 Thermo Scientific, Inc., Waltham, MA, USA). Samples of the hydrogels were freeze-dried for 48 h and triturated before being used, and then, 0.2 mg of dry sample and 10 mg of KBr were mixed, after mixture was pressured at a 700 y 1000 kg/cm^−1^and a disc was formed. The Fourier transform infrared (FT-IR) spectra were documented at a wavelength ranging from 4000 to 600 cm^−1^ with the 1 cm^−1^ resolution and recorded at 25 °C. Measurements were performed in triplicate per group.

### 2.7. Scanning Electron Microscopy (SEM)

The microstructure and morphology of Ge/HA scaffolds were visualized using a scanning electron microscopy (SEM) Model: Zeiss DSM950, from Zeiss (Oberkochen, Germany) at 10 kV acceleration voltages. First, the scaffolds were sputtered with gold using a Vacuum Sputtering and Thermal Coating System Polaron model 11HD (Hertfordshire, England for 1 min to yield approximately 10 nm thick film. Four SEM images were taken at 1500X for further analysis.

### 2.8. Melanoma Induction in B16-F10 Mice

For the induction of melanoma, 60,000 B16-F10 melanoma cells were inoculated subcutaneously into the abdominal region of C57BL/6 mice.

### 2.9. Effect of the Ge/HA Scaffold Coupled to CpG, MAGE-A5 or Both in Splenocytes In Vitro Tests

Splenocytes from C57BL/6 mice previously injected with melanoma cells 15 days earlier were incubated in RPMI 1460 medium (10% fetal bovine serum and 1% antibiotics) from Biowest (Nuaillé, France) for 5 days at 37 °C and 5% CO_2_ with the following scaffolds: (a) Ge/HA, (b) Ge/HA/CpG, (c) Ge/HA/MAGE-A5, and (d) Ge/HA/MAGE-A5/CpG. Subsequently, the activation of lymphocytes and antigen-presenting cells was analyzed. For the above, the cells were stained with the following antibodies, all at a dilution of 1:300: anti-Ia/Ie FITC, anti-CD3 FITC, anti-CD4 APC, anti-CD8 PercPE, anti-CD137-biotin, anti-CD11c APC, anti-CD40- PE, anti-CD86-PE, CD80 PE, and anti-F4/80 Cy5. The stained cells were acquired in a FACSCalibur cytometer at the National Laboratory of Flow Cytometry of the National Institute of Biomedical Research, UNAM, and analyzed in Flow Jo 8.7 software. Moreover, to analyze the interaction between cells and the scaffolds, Ge/HA scaffolds seeded with splenocytes were stained with hematoxylin to obtain photomicrographs with a Nikon camera (Tokyo, Japan) adapted to an Olympus IX71 inverted microscope Nikon (Tokyo, Japan).

### 2.10. Tumor Growth Rate and Survival

C57BL/6 mice were first inoculated with the hydrogels formed by the Ge/HA, Ge/HA/CpG, Ge/HA/MAGE-A5, and Ge/HA/CpG/MAGE-AX scaffolds; three weeks later, mice were challenged with 60,000 melanoma cells. Mice were monitored daily, and when the tumor lesion was visible. The development of the tumor lesions depends on each mouse and the treatment received. Untreated melanoma mice generally start to show the lesions after two weeks of tumor cell inoculation, while mice treated with the various scaffolds took about a week longer. The largest and smallest diameters were measured every two days using an electronic Vernier caliper. Then, the tumor volume was calculated using the formula V = (A2 × B)/2, where A corresponded to the smallest diameter and B to the largest tumor diameter.

### 2.11. Histopathological Evaluation of Tumors

For the histopathological evaluation of the tumor lesions, tumors were dissected and then fixed for 48 h in 10% formalin. The samples were washed, dehydrated, and embedded in paraffin. Subsequently, sections of approximately 7 µm were made and stained with hematoxylin and eosin for the histopathological study. The histological study was completed by taking micrographs showing a 1.9 mm^2^ area. The histological sections were photographed using a 20× objective on a Nikon Eclipse 80i microscope (Tokyo, Japan).

### 2.12. Statistical Analysis

A survival analysis was performed with the different groups of treated mice. For evaluation of the effect of the different treatments on the populations of macrophages, DC, and T lymphocytes and on the production of cytokines, ANOVA tests were carried out, and Tukey’s test was used to verify changes between groups. A *p* < 0.05 was considered significant. All analyses were performed in the GraphPad 9 program (San Diego, CA, USA).

## 3. Results

### 3.1. FT-IR Analysis

In the FT-IR spectra of the Ge/HA polymer, we found a strong band at 3400–3600 cm^−1^ belonging to stretching vibration of OH group in the Ge/HA without EDC, while this band was reduced in the Ge/HA with EDC, which means there was less water absorption due to crosslinking. Moreover, absorption bands at 1650, 1080, and 1614 were assigned to the C=O, C-H and COOH bonds corresponding to the structure of hyaluronic acid, as well as peaks at 1545 and 1455 cm^−1^ that evidence the N-H and C-C bounds, corresponding to the structure of the gelatin. All the peaks were presented in scaffolds with or without EDC (Figure 1).

### 3.2. Surface Morphology of Ge/HA

Analysis of the SEM imaging confirmed porous appearance with a mean diameter of 30.5 µm (Figure 2) as we described previously [26].

### 3.3. Effect of the Ge/HA Scaffold Coupled to CpG, MAGE-A5 or Both in Splenocytes In Vitro Tests

For determination of the effect of Ge/HA scaffolds on macrophages, DC, and T cells, splenocytes from C57BL/6 mice with melanoma were cultured with Ge/HA scaffolds coupled to CpG, MAGE-A5, or both for 5 days. The percentage of macrophages and DC and the expression of MHCII and the costimulatory molecules CD40, CD86, and CD80 were analyzed, as well as the percentage of CD4 and CD8 lymphocytes and the expression of the activation molecule CD137.

#### 3.3.1. Effect of Scaffolds on Macrophages and Dendritic Cells

Regarding the effect of the scaffolds on macrophages and DC, a slight increase in the percentage of DC was observed in the cultures treated with GE/HA and GE/HA/CpG compared to that of the WT group, while only the percentage of macrophages in the cells seeded with the Ge/HA scaffold increased slightly compared to that of the WT group. However, no significant changes were observed in the percentages of macrophages and DC (Figure 3).

To verify the degree of DC maturation and macrophage activation, we analyzed the expression of the CD40, CD80, and CD86 molecules. The most evident changes were observed with CD40. The increase in the expression of CD40 was evident in the DC among all the experimental groups. When the cells were seeded with the Ge/HA and Ge/HA/CpG scaffolds, CD40 expression increased in comparison with that of the control group, although when MAGE-A5 was coupled to the scaffolds, especially in CpG/MAGE-A5-treated cells, the expression increased markedly, and it was even higher than that induced by the Ge/HA scaffolds and those coupled to CpG (Table 1 and Figure 4). While the macrophages increased the expression of CD40 when they were seeded with the scaffolds coupled to CpG, the increased expression of CD40 was more notable in the cells treated with the MAGE-A5 and MAGE-A5/CpG scaffolds than in the cells of the WT, Ge/HA and Ge/HA/CpG groups (Table 1 and Figure 4). Notably, when DC and macrophages were treated with the scaffolds coupled to MAGE-A5 and the mixture MAGE-A5/CpG, CD40 levels were higher than the expression caused by Ge/HA and the CpG scaffolds.

However, in the case of CD80, a positive trend was observed in the expression in both cell types, since the Ge/HA scaffolds alone or coupled to CpG induced an increase in CD80 expression vs. the scaffolds coupled to MAGE-A5 and MAGE/A5/CpG; however, significant changes were only observed in the DC (Table 1 and Figure 4). When the effect of the scaffolds on CD86 was analyzed, the changes in both macrophages and DC were notable. In the case of DC, only Ge/HA scaffolds mediated increased CD86 expression in comparison to the WT, Ge/HA/MAGE, and Ge/HA/MAGE/CpG. In contrast, macrophages showed increased expression of CD86 when treated with Ge/HA/MAGE/CpG compared to that of the WT group and the Ge/HA/MAGE-A5 group. The Ge/HA and Ge/HA/MAGE groups also showed a slight increase in CD86 expression compared to the WT group; however, no significant changes were observed (Table 1 and Figure 4).

#### 3.3.2. Effect of Scaffolds on T Lymphocytes

To analyze whether the scaffolds could induce the activation of T lymphocytes, we determined the percentages of CD4 and CD8 T lymphocytes, in addition to the expression of the activation molecule CD137. A slight increase in the percentage of CD4 T cells was observed in the splenocytes from the Ge/HA/CpG, Ge/HA/MAGE-A5, and Ge/HA/CpG/MAGE-A5 groups vs. the Ge/HA and WT groups; nevertheless, the changes were not significant. In contrast, the percentage of CD8 T cells only showed changes in the Ge/HA-treated group of splenocytes in comparison with cells treated with Ge/HA/MAGE-A5 and Ge/HA/CpG/MAGE-A5 (Figure 5). Nonetheless, when CD137 expression was analyzed, changes were noted in both CD4+ and CD8+ T lymphocytes. CD4 T lymphocytes showed increased expression of CD137 in the groups treated with Ge/HA, Ge/HA/CpG, and Ge/HA/CpG/MAGE-A5 compared to the WT group. CD8+ lymphocytes showed the greatest increase in CD137 expression, especially in the groups treated with Ge/HA/MAGE-A5 and Ge/HA/CpG/MAGE-A5 compared to the WT, Ge/HA, and Ge/HA/CpG groups (Figure 5).

### 3.4. Survival Analysis and Tumor Growth Rate

For the analysis of tumor survival and growth, C57BL/6 mice were first treated with the Ge/HA, Ge/HA/CpG, Ge/HA/MAGE-A5, and Ge/HA/CpG/MAGE-A5 scaffolds. Three weeks later, mice were challenged with 60,000 cells of the B16-F10 melanoma cell line. Regarding survival, the group treated with Ge/HA/CpG showed the highest survival from the beginning of the treatment in comparison to all the experimental groups; although by day 28–34 post-melanoma inoculation, both the Ge/HA/CpG- and Ge/HA/MAGE-A5-treated mice reached 25% survival, much higher than that of the other groups. In contrast, the WT, Ge/HA, and Ge/HA/MAGE-A5/CpG-treated mice showed the same percentage survival during days 24 and 26 after melanoma inoculation (60–62% of survival), but at days 30–34 post-inoculation, the WT group showed 0% survival, the Ge/HA group 13%, and the Ge/HA/MAGE-A5/CpG Group 38% (Figure 6a).

The tumor growth rate was also analyzed. The group treated with Ge/HA/CpG showed the lowest tumor growth rate of all groups; even among the treated mice, only some developed tumor lesions. As a result, from day 28 post-inoculation, a tumor growth rate of zero was maintained. The group treated with Ge/HA/MAGE-A5 showed a similar behavior; however, the mice that had not developed visible tumor lesions at day 32 began to show them at day 36. The mice treated with Ge/HA and Ge/HA/MAGE-A5/CpG showed similar growth rates; nevertheless, it is evident that some of the mice treated with Ge/HA developed lesions much faster than other mice with the same treatment, showing a limited growth rate at day 28, which caused the loss of mice that developed an approximate tumor volume of 5–6 mm^3^. In the case of the group treated with Ge/HA/MAGE-A5/CpG, all mice developed tumor lesions from the beginning; however, the growth was not as fast as that achieved by the mice treated with Ge/HA. In contrast, the WT group achieved a much higher tumor volume than all experimental groups (Figure 6b).

### 3.5. Histopathological Evaluation of Tumor Lesions

To analyze the effect of therapy with Ge/HA scaffolds coupled to CpG and/or MAGE-A5 in mice with melanoma, we performed a histopathological study of the tumor parenchyma. In WT mice, the presence of elongated tumor cells with clear basophilic and eosinophilic regions, related to high cell activity, was confirmed. In some regions, the presence of tumor cells with abundant melanin granules was noted, indicating cell differentiation. Abundant capillaries and some unilocular adipocytes were also observed (Figure 6c). While the tumor parenchyma of the mice with melanoma treated with the Ge/Ha scaffolds was very similar to that of the mice without treatment, an increase in the presence of areas with cells with abundant melanin was noted (Figure 6d). After treatment with Ge/HA/CpG and Ge/HA/MAGE-A5, in addition to the presence of tumor cells with basophilic and acidophilic regions and abundant areas with pigmented cells, extensive eosinophilic areas formed by necrotic and apoptotic cells were evident (Figure 6e,f). Surprisingly, the mice treated with the Ge/HA/CpG/MAGE-A5 scaffolds showed more areas of active tumor cells and blood vessels than the representative eosinophilic areas of cell death seen in the mice treated with Ge/HA/CpG or Ge/HA/MAGE-A5 (Figure 6g).

## 4. Discussion

The effect of Ge/Ha scaffolds coupled to CpG and MAGE-A5 as a treatment against murine melanoma was analyzed. C57BL/6 mice were first treated with Ge/HA scaffolds coupled to MAGE-A5 and/or CpG and then challenged with B16-F10 melanoma cells. A histopathological analysis of the tumors was performed, including the survival, tumor growth rate, and immune response induced by the scaffolds. The groups treated with Ge/HA/CpG and Ge/HA/MAGE-A5 showed greater survival and a lower tumor growth rate, in addition to inducing DC maturation and macrophage activation. Regarding the tumor parenchyma, the mice with melanoma treated with Ge/HA/MAGE-A5 and Ge/HA/CpG displayed more areas with abundant tumor cells with melanin granules and cell death, which is related to the immune response developed, survival, and tumor growth rate. Only the Ge/HA/MAGE-A5 and Ge/HA/MAGE-A5/CpG scaffolds induced the activation of CD8 T cells. In conclusion, Ge/HA scaffolds coupled to CpG or MAGE-A5, but not the mixture, can induce a successful immune response capable of promoting tumor cell clearance and increased survival.

In the field of cancer immunotherapy, the search for strategies or materials that may induce the development of immune responses that allow the elimination of tumor lesions is ongoing. For years, adjuvants have enabled immune responses that small molecules, such as antitumor peptides, cannot achieve [27]. Currently, not only the effect of adjuvants is analyzed but also that induced by scaffolds made of biomaterials. Some of these can be recognized by the immune system, promoting the migration and adhesion of leukocytes, and even protect molecules coupled to the scaffolds from degradation [27]. Most of the materials used in immunotherapy have a synthetic origin, so some must be functionalized to stimulate the desired immune responses, while others can modulate the immune system [27]. For example, polylactic-co-glycolic acid (PLGA) can be recognized and then phagocytosed by DC [28]. Other work with PLG also showed the induction in the migration of DC [25]. Another example is polyanhydric compounds such as 1,6-bis(p-carboxyphenoxy) hexane (CPH), which induce DC maturation in vitro [29]. Biomaterials such as alginate, fibrin, and silica have also been used for scaffold construction [8,28,29,30]; nevertheless, only the alginate scaffolds induced DC migration [30], while the others increased the survival of mice with tumor lesions after they were coupled to CpG, GM-CSF, DC, or TCAR lymphocytes [31,32].

In the present project, a Ge/HA scaffold coupled to CpG and MAGE-A5 was constructed. Compared to the scaffolds, Ge/HA alone was shown to induce several responses in mice with melanoma. First, Ge/HA scaffolds increased survival and decreased the tumor growth rate in comparison with those of the melanoma cell-challenged mice (Figure 6a,b); in addition, GE/HA scaffolds mediated the increase in costimulatory molecule expression in macrophages and DC and a slight increase in the percentage of CD8 T lymphocytes and the activation of CD4 T lymphocytes, in contrast to what was observed in the mice with melanoma without treatment (Figure 5). Both Ge and HA may be involved in the induction of leukocyte migration. Ge is a derivative of collagen, a very abundant protein in the extracellular matrix with multiple cell adhesion motifs [10]. Ge possesses RGD sites that allow leukocyte adhesion [13], in addition to being involved in the production of low levels of TNFα [10]. While HA induces leukocyte migration and adhesion through the interaction of CD44 and CD168, HA is also recognized by Toll-like receptors 2 and 4 by APCs, especially by DC. Macrophages can be activated through the recognition of HA by interaction with the HA phagocytosis receptor (HARE) [11]. Thus, it is possible that the microenvironment formed by both Ge and HA in the scaffold allowed the maturation of the APCs, contributing to the subsequent activation of T lymphocytes.

In addition, the fact that the molecular composition of Ge and HA in the scaffolds was preserved after EDC crosslinking, as observed by TF-IR, allowed DC and macrophages to interact with the scaffolds to be stimulated and overexpress class II and CD40 molecules. Figure 2 shows close contacts between the DC and macrophages to Ge/HA scaffolds. It is well known that DC and macrophages have adhesion molecules towards extracellular matrix proteins as mentioned above. On the other hand, the fact that the Ge/HA scaffolds were porous allowed DC penetration, ensuring their good stimulation. The above facts showed that the molecular and topographical characteristic of Ge/AH scaffolds played an important role in the functional behavior and activation of the immune response in mice with melanoma promoting the destruction of tumor cells.

In addition to the above, the adjuvant effect of Ge/HA increased the effect induced by CpG and MAGE-A5 in the immune response, as evidenced by the survival and tumor growth rate assays (Figure 6). Notably, the mice treated with the Ge/HA/MAGE-A5 scaffolds and especially with Ge/HA/CpG showed the highest survival, the lowest tumor growth rate, and the presence of abundant cell death zones in the tumor parenchyma. In the case of the Ge/HA/MAGE-A5 scaffold, the activation of an antigen-specific antitumor immune response was clearly induced (Figure 5). The MAGE-A5 peptide is presented via H2-Kb, which is why it is involved in the activation of CD8 T lymphocytes [24], which are cells related to the elimination of cells expressing tumor-specific antigens. The Ge/HA/MAGE-A5 scaffold, despite not showing changes in the percentage of CD4 and CD8 T lymphocytes, did induce the expression of the activation molecule CD137 in CD8 T lymphocytes (Figure 5). Clearly, the addition of MAGE-A5 to the Ge/HA scaffold increased survival and decreased the tumor growth rate and the appearance of abundant cell death zones in the tumor parenchyma compared to those of the WT group and the Ge/HA-treated group. In addition to the above, for CD8 T lymphocytes to be activated, the maturation and activation of antigen-presenting cells are essential. Notably, the Ge/HA/MAGE-A5 scaffold induced an increase in the expression of CD40 in DC and CD40 and CD86 in macrophages, relevant molecules during the presentation of antigens to T lymphocytes. The scaffolds did not contain cells or chemokines involved in the migration of APCs or T lymphocytes, so the microenvironment formed by the Ge/HA scaffold promoted the migration and activation of APCs in situ, and MAGE-A5 promoted an antigen-specific antitumor immune response. The MAGE-A5 peptide had already been used in immunotherapy with DC from the spleen and incubated with the antigen [24]. In DC-based immunotherapy, approximately 1% of the inoculated cells reach the lymph nodes [33], so the activation of APCs in situ is important, especially because the cells may migrate to the scaffold and then activate. In contrast, in the case of immunotherapy based on DC, there are multiple costs to induce differentiation and subsequent activation of cells, in addition to the loss of cell viability. Furthermore, on several occasions, patients must receive several inoculations to develop an antigen-specific immune response. In the present investigation, the results suggest that the implantation of the scaffold did not allow the formation of a microenvironment that prompted the activation of APCs and CD8 T lymphocytes, which are essential cells in the antitumor immune response [34].

The Ge/HA scaffold coupled to CpG resulted in the best survival, the lowest tumor growth rate, and the highest expression of costimulatory molecules in the DC. The unmethylated cytosine-guanine dinucleotide (CpG) motifs are sequences found in the vesicles of prokaryotes [35]. CpG are considered PAMPs, as they are recognized by TLR9 and preferentially expressed in DC, inducing the production of IFNα, which can promote the development of a Th1 response. Additionally, CpG stimulate the activation of B and NK cells [15]. Due to the CpG effects in the immune system, they have been used in immunotherapy against cancer, asthma, and other diseases [17]. Regarding antitumor therapy, CpG in conjunction with an anti-TLR2 antibody have been shown to induce the production of IFNγ and the migration of CD8 T and NK lymphocytes to the tumor stroma [36,37]. In addition, CpG have been coupled to PLG or graphene-chitosan scaffolds with GM-CSF, stimulating increased survival of mice, activation of CD8 T lymphocytes and IFNγ production [25,30]. The effects observed after the application of CpG are promising; however, it is important to note that CpG have been used in conjunction with other molecules that exacerbate their effect. First, an anti-TLR2 antibody was used to synergize the effect induced by the recognition of CpG by TLR9, whereas when CpG were coupled to the materials, GM-CSF was used to induce migration of APCs to the scaffold, thus promoting CpG recognition. In the present project, both HA and Ge appeared to be contributors to the effects induced by CpG, due to the effects that these biomaterials have on the migration, adhesion and activation of leukocytes [10,11,13]. Therefore, in this case, CpG coupled to the scaffolds showed that the innate immune response was related to the appearance of large areas of cell death in tumor lesions, in the decrease in the tumor growth rate, in the increase in survival, in the maturation of DC and in the activation of macrophages, although the activation of T lymphocytes was not confirmed. In this sense, it is very likely that NK lymphocytes have played a relevant role, since they are cells of the innate immunity involved in antitumor immunosurveillance to eliminate the tumor cells [34]. Thus, it is possible that the Ge/HA/CpG scaffold induced NK cell activation through the interaction of CpG with DC, mediating their maturation and subsequent activation of NK cells through IFNα production [38].

Finally, Ge/HA/CpG/MAGE-A5 scaffolds were also constructed and found to induce the greatest protection against melanoma. The results showed a higher survival and lower tumor growth rate than that produced by Ge/HA scaffolds but not a higher rate than that caused by Ge/HA/CpG or Ge/HA/MAGE-A5 (Figure 6). Notably, this group showed only an increase in CD40 in both macrophages and DC but not in the other costimulatory molecules. However, the effect on the activation molecule CD137 in CD8 T lymphocytes was important. The results of the scaffold were not as expected, since it mediated lower effects than those induced by MAGE-A5 or CpG when they were coupled individually to the scaffolds. Previously, work by Donaldson in 2018, in which photocrosslinked Ge scaffolds were constructed, showed that the scaffolds could promote TNFα production in human monocytes, but when monocytes were activated with LPS, the scaffold provoked decreased production of TNFα. In this sense, it is likely that Ge prevents the uncontrolled exacerbation of inflammation, thus preventing the development of the response that could be mediated by the CpG/MAGE-A5 mixture.

## 5. Conclusions

Ge/HA scaffolds by themselves caused an increase in survival and a decrease in the rate of melanoma growth. The most successful response was shown by Ge/HA scaffolds coupled to MAGE-A5 or CpG, which induced DC maturation, macrophage activation, increased survival, and decreased the tumor growth rate, in addition to the appearance of multiple areas of cell death in the tumor parenchyma. In the case of the Ge/HA/MAGE-A5 scaffolds, an antigen-specific immune response dependent on CD8 T lymphocytes was produced, while the Ge/HA/CpG scaffolds produced an innate immune response. Therefore, the Ge/HA scaffolds may be successful in tumor therapy since they help to increase the immune response induced by molecules such as CpG or MAGE-A5 but prevent exacerbated immune responses that may have toxic effects

## Figures and Tables

**Figure 1 polymers-14-04608-f001:**
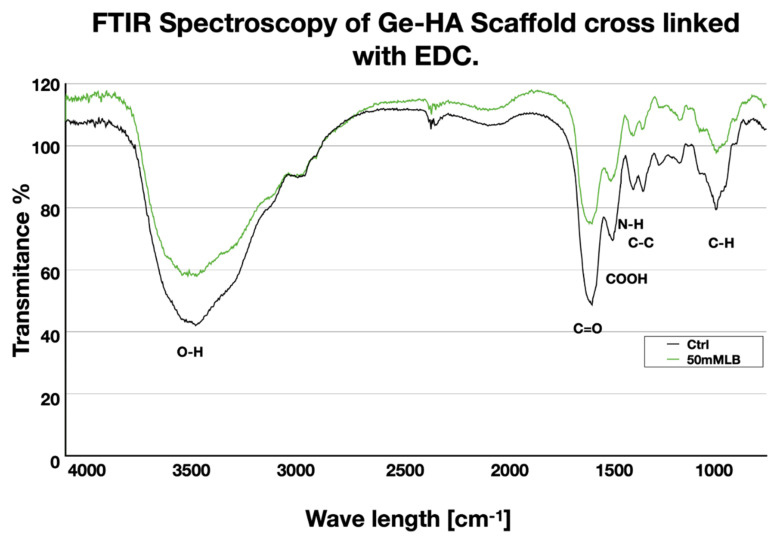
FT-IR spectra of Ge/HA with EDC and Ge/HA without EDC. It can be seen that both scaffolds have a similar molecular composition.

**Figure 2 polymers-14-04608-f002:**
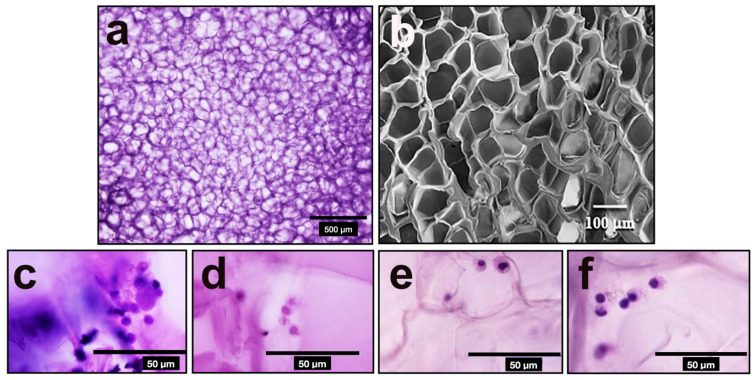
Morphology of Ge/HA scaffolds intertwined with EDC: (**a**) paraffin section of a scaffold without cells stained with H-E. (**b**) SEM of Ge/HA without cells; (**c**,**d**) paraffin sections of total spleen cell culture on Ge/HA scaffolds stained with H-E; (**e**,**f**) paraffin sections of splenic DC cultured on Ge/HA scaffolds stained with H-E. In (**a**,**b**), it can be seen that the scaffolds are porous and that they are interconnected. Furthermore, in (**c**,**d**), it can be seen that the DC are closely adhered to the trabeculae of the Ge/HA scaffolds.

**Figure 3 polymers-14-04608-f003:**
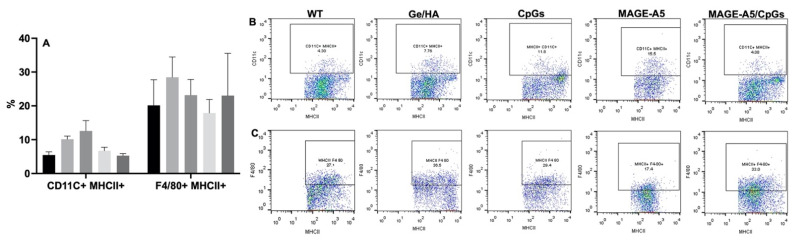
Percentage of macrophages and DC after treatment with Ge/HA scaffolds coupled to CpG or MAGE-A5: (**A**) graph of the percentage of DC and macrophages; (**B**) representative dot blots of the percentage of DC; (**C**) dot blots representative of the percentage of macrophages. 
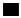
 Without treatment, 
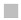
 Ge/Ha, 
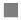
 CpG, 
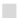
 MAGE-A5, 
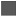
 MAGE-A5/CpG.

**Figure 4 polymers-14-04608-f004:**
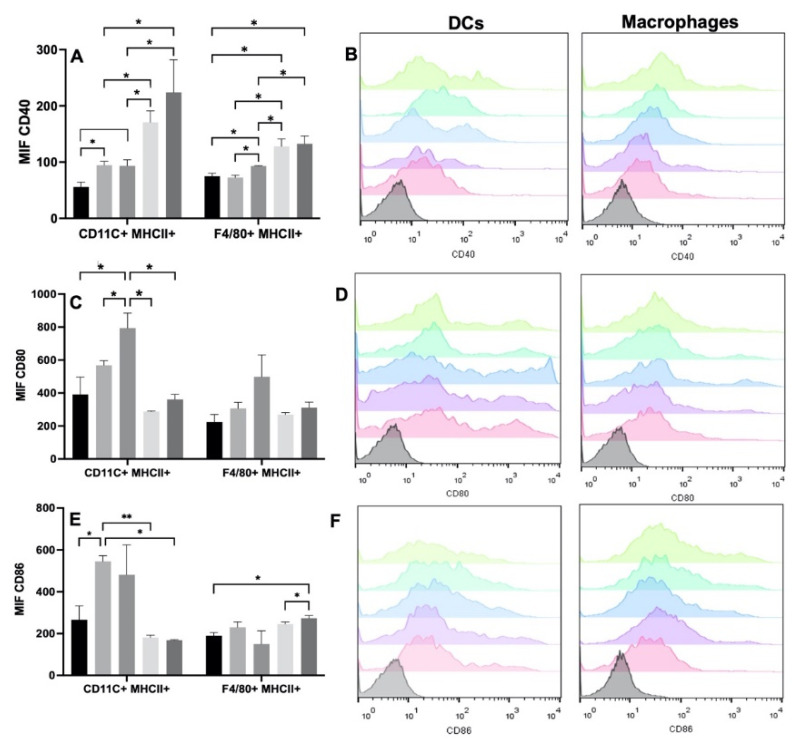
Expression of costimulatory molecules in DC (CD11c+ MHCII+) and macrophages (F4/80+ MHCII+) after treatment with Ge/HA scaffolds coupled to MAGE/A5 or CpG. Without treatment, Ge/Ha, CpG, MAGE-A5, MAGE-A5/CpG. (**A**) CD40 expression. (**B**) Representative histograms of CD40 expression in DC and macrophages. (**C**) CD80 expression. (**D**) Representative histograms of CD80 expression in DC and macrophages. (**E**) Expression of CD86. (**F**) Representative histograms of CD86 expression in DC and macrophages. No significant changes were found in the levels of macrophages and DC after the treatments. Regarding costimulatory molecules, the increase in CD40 expression was evident in DC among all experimental groups (* *p* = 0.012: Ge/HA vs. WT, Ge/HA/MAGE-A5, Ge/HA/MAGE-A5/CpG; Ge/HA/CpG vs. Ge/HA/MAGE-A5 and Ge/HA/MAGE-A5/CpG). Macrophages also showed increased CD40 expression in the groups treated with Ge/HA/CpG, Ge/HA/MAGE-A5, and Ge/HA/CpG/MAGE-A5 compared to the WT group and among the other experimental groups (* *p* = 0.0125: Ge/HA/CpG vs. WT, Ge/HA, Ge/HA/MAGE-A5, Ge/HA/MAGE-A5/CpG, Ge/HA/MAGE-A5 vs. WT, Ge/HA, Ge/HA/CpG, Ge/HA/CpG/MAGE-A5). In the case of CD80 in DC, Ge/HA scaffolds alone or coupled to CpG induced increased CD80 expression vs. cells treated with the other groups (* *p* = 0.013, Ge/HA/CpG vs. Ge/HA, * *p* = 0.013 Ge/HA/CpG vs. Ge/HA/MAGE-A5 and Ge/HA/MAGE-A5/CpG). When the effect of scaffolds on CD86 was analyzed, only Ge/HA scaffolds induced increased expression in DC compared to WT, Ge/HA/MAGE-A5 and Ge/HA/MAGE-A5/CpG (** *p* = 0.006: Ge/HA vs. Ge/HA/MAGE-A5; * *p* = 0.012: Ge/HA vs. WT and Ge/HA/MAGE-A5/CpG). In contrast, macrophages showed increased expression of CD86 when treated with Ge/HA/MAGE/CpG compared to the WT group and Ge/HA/MAGE-A5 (* *p* = 0.04: Ge/HA/MAGE-A5/CpG vs. WT, Ge/HA/MAGE).

**Figure 5 polymers-14-04608-f005:**
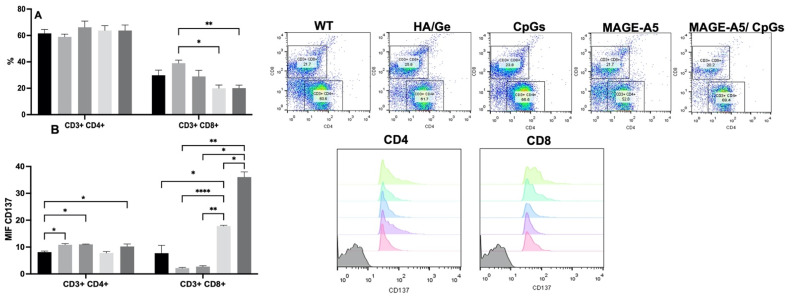
Percentage of T lymphocytes and expression of CD137 after treatment with Ge/HA scaffolds coupled to CpG and/or MAGE-A5. (**A**) Percentage of CD4 T lymphocytes. 
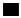
 Without treatment, 
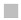
 Ge/Ha, 
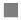
 CpG, 
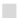
 MAGE-A5, 
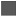
 MAGE-A5/CpG. (**B**) Expression of CD137 in CD4 and CD8 T lymphocytes. No significant changes were observed in the percentage of CD4 T lymphocytes. CD8 T lymphocytes only showed an increase in the Ge/HA group compared to the Ge/HA/MAGE-A5 and Ge/HA/CpG/MAGE-A5 groups (* *p* = 0.01: Ge/HA vs. Ge/HA/MAGE-A5; ** *p* = 0.003: Ge/HA vs. Ge/HA/MAGE-A5/CpG). CD137 expression was enhanced in both CD4+ and CD8+ T lymphocytes. CD4 T lymphocytes showed higher levels of CD137 in the groups treated with Ge/HA, Ge/HA/CpG, and Ge/HA/CpG/MAGE-A5 than in the WT group (* *p* = 0.03: WT vs. Ge/HA, Ge/HA/CpG, Ge/HA/MAGE-A5/CpG). CD8+ lymphocytes showed the highest CD137 expression in the Ge/HA/MAGE-A5- and Ge/HA/CpG/MAGE-A5-treated groups compared to the WT groups (* *p* = 0.04 WT vs. Ge/HA/MAGE-A5; * *p* = 0.02: Ge/HA vs. Ge/HA/MAGE-A5/CpG; * *p* = 0.01: Ge/HA/CpG vs. Ge/HA/MAGE-A5/CpG; ** *p* = 0.005: Ge/HA/CpG vs. Ge/HA/MAGE-A5; ** *p* = Ge/HA vs. Ge/HA/MAGE-A5/CpG; **** *p* < 0.0001: Ge/HA vs. Ge/HA/MAGE-A5).

**Figure 6 polymers-14-04608-f006:**
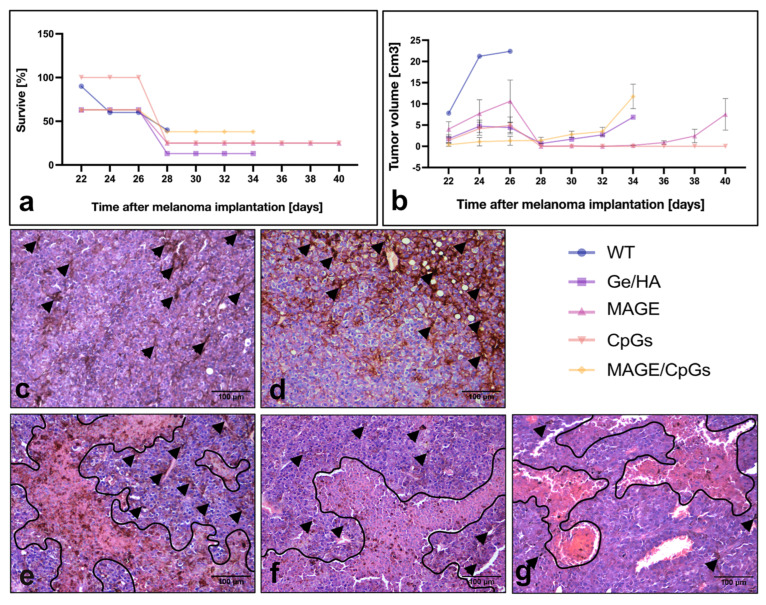
Histopathological study of the tumor parenchyma, survival, and tumor growth rate of the mice with melanoma treated with Ge/HA coupled to CpG or MAGE-A5: (**a**) survival and (**b**) tumor growth rate of the mice with melanoma treated with Ge/HA scaffolds coupled to MAGE-A5 and/or CpG; (**c**) without treatment (WT); (**d**) Ge/HA; (**e**) CpG; (**f**) MAGE-A5; (**g**) MAGE-A5/CpG. WT mice possessed a tumor stroma made up of abundant viable and active cells with basophilic staining and a few acidophilic regions typical of areas with abundant nonviable cells. The CpG and MAGE-A5 groups had tumor stroma with some basophilic viable cell areas and abundant acidophilic regions typical of cell death areas. Scaffolds coupled to CpG and MAGE-A5 contained both acidophilic and basophilic regions, in addition to cells with melanin (arrowheads). The groups treated with the scaffolds coupled to CpG or MAGE-A5 showed the lowest tumor growth rate and the highest percentage of survival. The areas of necrosis in histological sections were outlined (black line) to better visualization.

**Table 1 polymers-14-04608-t001:** Expression of costimulatory molecules in DC and macrophages. The expression of each costimulatory molecule is represented by the average of the mean fluorescence intensity (MIF) plus standard deviation (SD). *p* < 0.05 was considered significant. dna: does not apply; ns: not significance.

Experimental Group	CD40	CD80	CD86	*p*
Without treatment (WT)	DC: 56 ± 8.4	DC: 391 ± 105.65	DC: 266.5 ± 65.9	dna
	MO: 74.9 ± 5.1	MO: 224 ± 45.6	MO: 189.5 ± 15.3	dna
Ge/HA	DC: 94.6 ± 6.58	DC: 567 ± 28.57	DC:245.5 ± 26.9	CD40: *p* = 0.012. Ge/HA vs. WT, MAGE-A5 and MAGE-A5/CpG CD86: *p* =0.006. Ge/HA vs. MAGE-A5. *p* = 0.012. Ge/HA vs. WT and MAGE-A5/CpG
	MO:72.75 ± 4.07	MO:307.5 ± 35.5	MO: 230 ± 25.9	ns
MAGE-A5	DC: 170 ± 20.5	DC: 287.5 ± 3.7	DC: 181 ± 10.9	ns
	MO: 128 ± 13.3	MO: 269 ± 12.17	MO: 246 ± 9.8	CD40: *p* = 0.0125. MAGE-A5 vs. WT, Ge/HA, CpG and MAGE-A5/CpG
CpG	DC: 93.3 ± 10.8	DC: 795 ± 91.2	DC: 480.5 ± 142.9	CD40: *p* = 0.012. CpG vs. MAGE-A5 and MAGE-A5/CpG CD80: *p* = 0.013. CpG vs. Ge/HA and MAGE-A5
	MO:3.05 ± 0.7	MO: 499 ± 132.2	MO: 150 ± 62.2	CD40: *p* = 0.0125. CpG vs. WT, Ge/HA, MAGE-A5, MAGE-A5/CpG.
MAGE-A5/CpG	DC: 224 ± 57.7	MO: 361 ± 30.6	DC: 168 ± 288	ns
	MO: 132 ± 14.1	MO: 311.5 ± 33.19	MO: 273 ± 13.2	CD40: *p* = 0.0125. CpG/MAGE-A5 vs. WT CD86: *p* = 0.04. MAGE-A5/CpG vs. WT and MAGE-A5.

## Data Availability

The data presented in this study are available on request from the corresponding author.

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
