# Peer review of "Gelatin/Hyaluronic Acid Scaffold Coupled to CpG and MAGE-A5 as a Treatment against Murine Melanoma"

_polymers, 2022, doi:10.3390/polym14214608_

Round 1

Reviewer 1 Report

Dear authors

In my opinion, the importance of the topic is clear to the biomaterials community, however, there are some limitations in the work, which should be addressed by the authors before further consideration.

1- The language aspects of the manuscript need to be improved.

2- The last paragraph of the introduction needs to be modified.

3- In the experiment, please add more details for raw materials. For example, the molecular weight, manufacturer, etc. of the polymers, GA and HA, were missed.

4- The experimental section has to be modified completely and more data should be provided. For example, the power of sonication in scaffold fabrication? The manufacturer of raw materials? Time of the process? ….

5- Would you provide the characterization data (e.g., SEM and FTIR) of the fabricated polymers?

6- If possible, improv and change some parts of figure 4.

Good luck

Author Response

Referee 1

Dear authors

In my opinion, the importance of the topic is clear to the biomaterials community, however, there are some limitations in the work, which should be addressed by the authors before further consideration.

1- The language aspects of the manuscript need to be improved.

Thanks for your suggestion. A native English-speaking copyeditor reviewed the entire document. We add the certification that supports it.

2- The last paragraph of the introduction needs to be modified.

We thank the referee for his kind suggestion. We have changed the last paragraph of the introduction as follow inlines 80-85:

Thus, scaffolds built with extracellular matrix biomaterials can be an alternative to improve the results obtained by classical cancer immunotherapy [22,23], since they can protect immunomodulatory molecules from degradation and allow their gradual release. In this sense, CpG and MAGE antigens coupled to a Ge and HA scaffold will likely trigger a more robust antitumor immune response. Thus, the objective of this study was to evaluate this immune response in a murine melanoma model.

3- In the experiment, please add more details for raw materials. For example, the molecular weight, manufacturer, etc. of the polymers, GA and HA, were missed.

Thank you very much for mentioning this detail that we omitted. We have added the characteristics of Ge and HA in the material and methods. The added paragraph was as follows in lines 145-148:

Ge from bovine skin (Sigma - G9391) with a protein percentage of 78% and a water content of 8% was used. HA sodium salt from Streptococcus equi (Sigma - 53747) with a solubility of 5 mg/ml and an MW of ~1.5-1.8 x 10E6 Da was used.

4- The experimental section has to be modified completely and more data should be provided. For example, the power of sonication in scaffold fabrication? The manufacturer of raw materials? Time of the process? ….

Thanks for your suggestion. We have added several details in the manufacturing process of the scaffolding as follows:

  1. Lines 89-126:

To calculate the sample size, an equation based on the assessment of incidences was used:

X= N((A/100) x (B/100) x (Cx100))

where

X= Final number of animals needed

N= is the minimum statistical number that allows to conclude the objectives proposed in the project

A= 100- % incidence 1

B= 100-% incidence 2

C= 100-% incidence 3 and so on

In our project we have the following data:

N= at least 5 mice according to various articles in which melanoma induction is performed.

A= 10. Probability of death of a mouse not attributable to the procedure.

B= 10. Probability of death due to some complication in the procedure.

Substituting values:

X= 5/((90/100) (90/100))= 6.17 mice. Therefore, the model is adjusted to 6 mice per group.

  1. b. Lines 153-154

(IKA C-Mag HS7) at 750 rpm

  1. Lines 156-157

according to Eggert 2004 and Ali et al. 2008.

  1. Lines 163-163

with a Ultraturrax T25 (IKA) and 12,200 rpm for 1 min

  1. Line 163

Change @ to µ

  1. Lines 165-172

2.6 Fourier transform infrared (FT-IR) characterization

The chemical composition of Ge/HA polymers crosslinked with and without EDC were assessed by using a spectrometer (NICOLET Nexus 670 Thermo Scientific, Inc., Waltham, MA, USA). The Fourier transform infrared (FT-IR) spectra were documented at a wavelength ranging from 4,000 to 600 cm-1 with the 1 cm -1 resolution. Measurements were performed in triplicate per group.

  1. Lines 174-179

2.7. Scanning electron microscopy (SEM)

The microstructure and morphology of Ge/HA scaffolds were visualized using a scanning electron microscopy (SEM) Model: ZEISS DSM950 at 10kV acceleration voltajes. First, the scaffolds were sputtered with gold using a Vacuum Sputtering and Thermal Coating System (Polaron model 11HD) for 1 minute to yield approximately 10 nm thick film. Four SEM images were taken at 1500X for further analysis.

  1. Lines 180-182

2.8. Melanoma induction in B16-F10 mice

For the induction of melanoma, 60,000 B16-F10 melanoma cells were inoculated subcutaneously into the abdominal region of C57BL/6 mice.

  1. Lines 185-186

previously injected with melanoma cells 15 days earlier

  1. Line 183

change 2.7 to 2.9

  1. Line 190

, all at a dilution of 1:300:

  1. Lines 194-197

Also, to analyze the interaction between cells and the scaffolds, Ge/HA scaffolds seeded with splenocytes were stained with hematoxylin to obtain photomicrographs with a Nikon camera adapted to an Olympus IX71 inverted microscope Nikon.

  1. Line 198

change 2.8 to 2.10

  1. Lines 202-205

The development of the tumor lesions depends on each mouse and the treatment received. Untreated melanoma mice generally start to show the lesions after two weeks of tumor cell inoculation, while mice treated with the various scaffolds took about a week longer. Th

  1. Line 209

change 2.9 to 2.11

  1. Line 239

change 210 to 212

5- Would you provide the characterization data (e.g., SEM and FTIR) of the fabricated polymers?

The characterization of this scaffold was previously published. However, we have included FT-IR results of the Ge/HA scaffolds with and without EDC crosslinking (fig 1). We also include a SEM photomicrograph and several histological sections of scaffolds in Figure 2.

6- If possible, improv and change some parts of figure 4.

We have changed figure 4, now figure 6, outlining with black lines the areas of necrosis of the tumors to clarify the figures. In addition, the definition of the images was increased. We have also modified one of the charts where the % of cell data was missing.

Reviewer 2 Report

The objective of this study is to observe the efficacy of GE/HA scaffold coupled to CpG and MAGE antigens to be used as an immunotherapy treatment against experimental melanoma. However, in my opinion, this topic is not quite related to this journal "Polymers".

1. The experimental design was aimed to proof the efficacy. There are no the Ge/Ha scaffold interaction or any physicochemical properties of the scaffold.

2. The in vivo experiments are deep down in molecular effect which did not mention about the polymeric interaction.

In my opinion, this manuscript should submit or transfer to other journal such as "Molecules" (mdpi publication) 

I have a few questions based on the review.

1. The author should provide the sample size calculation. How did the author allocate 6 mice per group?

2. Topic 2.1 Mice, the author wrote six male C57BL/6 mice. Please explain why the authors used only 6 mice or 6 mice per group?

3. Do you have the control group as normal mice?

4. For the Table 1, please provide the significance of the comparison. (p value) The author mentioned that there are no significantly different from WT group on CD86 expression. How about there expression?

Author Response

Referee 2

The objective of this study is to observe the efficacy of GE/HA scaffold coupled to CpG and MAGE antigens to be used as an immunotherapy treatment against experimental melanoma. However, in my opinion, this topic is not quite related to this journal "Polymers".

  1. The experimental design was aimed to proof the efficacy. There are no the Ge/Ha scaffold interaction or any physicochemical properties of the scaffold.

Thanks for your suggestion. We add figure 2, where there is the interaction between the scaffolds and the DC. Also, we add a paragraph in the discussion on lines 628-637:

Besides, the fact that the molecular composition of Ge and HA in the scaffolds was preserved after EDC crosslinking, as observed by TF-IR, allowed DC and macrophages to interact with the scaffolds to be stimulated and overexpress class II and CD40 molecules. Figure 2 shows close contacts between the DC and macrophages to Ge/HA scaffolds. It is well known that DC and macrophages have adhesion molecules towards extracellular matrix proteins as mentioned above. On the other hand, the fact that the Ge/HA scaffolds were porous allowed DCs penetrating, ensuring its good stimulation. The above facts showed that the molecular and topographical characteristic of Ge/AH scaffolds played an important role in the functional behavior and activation of the immune response in mice with melanoma promoting the destruction of tumor cells.

  1. The in vivo experiments are deep down in molecular effect which did not mention about the polymeric interaction.

Thank you for your comment. As we have already mentioned, we have added a paragraph in discussion with what we consider clarifies the role of the Ge/Ha scaffold.

In my opinion, this manuscript should submit or transfer to other journal such as "Molecules" (mdpi publication) 

I have a few questions based on the review.

  1. The author should provide the sample size calculation. How did the author allocate 6 mice per group?
  2. Topic 2.1 Mice, the author wrote six male C57BL/6 mice. Please explain why the authors used only 6 mice or 6 mice per group?

Thank you for pointing out this omission. We have placed in the material and methods in the lines 89-126 the formula that we use to calculate the size of the groups of rats used in each experiment:

To calculate the sample size, an equation based on the assessment of incidences was used:

X= N((A/100) x (B/100) x (Cx100))

where

X= Final number of animals needed

N= is the minimum statistical number that allows to conclude the objectives proposed in the project

A= 100- % incidence 1

B= 100-% incidence 2

C= 100-% incidence 3 and so on

In our project we have the following data:

N= at least 5 mice according to various articles in which melanoma induction is performed.

A= 10. Probability of death of a mouse not attributable to the procedure.

B= 10. Probability of death due to some complication in the procedure.

Substituting values:

X= 5/((90/100) (90/100))= 6.17 mice. Therefore, the model is adjusted to 6 mice per group.

  1. Do you have the control group as normal mice? We considered the control group the mice with melanoma that did not receive any treatment.

Yes, like you, we consider mice with melanoma without any treatment to be the control group. In the experimental design, we do not have normal mice (without melanoma) that have received the scaffolds.

  1. For the Table 1, please provide the significance of the comparison. (p value) The author mentioned that there are no significantly different from WT group on CD86 expression. How about there expression?

Thanks for your observation. In Table 1, we have added the significance of each group.

Round 2

Reviewer 1 Report

Thank you. Now the work is acceptable for publication.

Author Response

Thank you very much for your comments that have improved this article.

Reviewer 2 Report

The authors provided all related information to fit in the Polymers journal which I have agreed that this manuscript should be published after minor revision.

1. In topic 2.6 FTIR, the author should provide more detail of the experiment such as the probe of detection or KBr compression?

2. Moreover, please check the unit in both text and figure 1 (cm -1 should be as exponent (-1)) and the transmittance should be in %T?

Author Response

Many thanks for your comments.

  1. Thanks for your observation.
    We have added the following paragraph in point 2.6, lines 148-150: Samples of the hydrogels were freeze-dried for 48 h and triturated before being used, then 0.2 mg of dry sample and 10 mg of KBr were mixed, after mixture was pressured at 700 and 1000 kg/cm-1 and a disc was formed.
    We also add in line 152 the following phrase: and recorded at 25°C.
  2. Thank you very much for your support for figure 1 to be improved. We have changed figure 1, where we have placed the exponent -1, and the transmittance was expressed in %.